# Transcriptomic Analysis Reveals C-C Motif Chemokine Receptor 1 as a Critical Pathogenic Hub Linking Sjögren’s Syndrome and Periodontitis

**DOI:** 10.3390/cimb47070523

**Published:** 2025-07-07

**Authors:** Yanjun Lin, Jingjing Su, Shupin Tang, Jun Jiang, Wenwei Wei, Jiang Chen, Dong Wu

**Affiliations:** 1Fujian Key Laboratory of Oral Diseases, School and Hospital of Stomatology, Fujian Medical University, Fuzhou 350004, China; yanjunlin@fjmu.edu.cn (Y.L.); jingjingsu2020@163.com (J.S.); wsjjwps10@sina.com (J.J.); weiwenwei@fjmu.edu.cn (W.W.); 2Department of Otorhinolaryngology-Head and Neck Surgery, The First Affiliated Hospital, Fujian Medical University, Fuzhou 350004, China; tangshupin@fjmu.edu.cn

**Keywords:** Sjögren’s syndrome (SS), periodontitis (PD), common genes, immune cell infiltration, C-C motif chemokine receptor 1 (*CCR1*)

## Abstract

Compelling evidence has demonstrated a bidirectional relationship between Sjögren’s syndrome (SS) and periodontitis (PD). Nevertheless, the underlying mechanisms driving their co-occurrence remain unclear, highlighting the need for finding the hub gene. This study sought to examine the common genes and any connections between SS and PD. Differently expressed genes (DEGs) were analyzed by means of gene set enrichment analysis (GSEA), weighted gene co-expression network analysis (WGCNA), and least absolute shrinkage and selection operator (LASSO) methods. The test and validation sets were used to depict the receiver operating characteristic (ROC) curves. The immune cell infiltration was performed via the cell-type identification by estimating relative subsets of RNA transcripts (CIBERSORT) methodology. The relationships between immune infiltrating cells and the common gene were examined. Ninety-five common genes with similar expression trends were obtained after DEGs analysis, which were enriched in cytokine—cytokine receptor interaction, chemokine signaling pathway, proteasome, intestinal immune network for IgA production, and cytosolic DNA sensing pathway. Thirty-nine common genes were obtained after WGCNA. Sixteen shared genes of DEGs analysis and WGCNA were incorporated into the LASSO model to obtain the unique shared gene, C-C motif chemokine receptor 1 (*CCR1*), which overexpressed and owned predictable ROC curves in test and validation sets. The examination of immune cell infiltration underscored its crucial roles in the disturbance of immune homeostasis and the emergence of pathogenic circumstances with the simultaneous occurrence of SS and PD. *CCR1* overexpresses and serves as a critical pathogenic hub linking SS and PD, which may play a role through immune cell infiltration.

## 1. Introduction

Sjögren’s syndrome (SS) is a progressively developing autoimmune illness primarily characterized by xerostomia and xerophthalmia. It is marked by diffuse lymphocytic infiltration of exocrine glands and other tissues, so SS can also be named autoimmune epitheliitis [1]. Autoantibodies targeting La and Ro proteins are often detected in patients with SS [2]. The overall prevalence rate of SS is 60.82 cases per 100,000 inhabitants with a strong female predilection (female/male ratio = 10.72) [3]. Nonetheless, the pathophysiology and etiology of SS remain inadequately comprehended. Genome-wide association studies have revealed genetic connections between SS and genes associated with both innate and adaptive immunological mechanisms [4]. Individuals with SS are prone to multi-organ failure. Various local chronic conditions, such as interstitial pneumonia and glomerulonephritis, are linked to increased SS levels [5].

Periodontitis (PD) is a chronic inflammatory illness with multiple contributing factors. The subgingival dental biofilm initiates a host inflammatory and immunological response, resulting in irreparable damage to the periodontium [6]. The specific pathogens and abnormal host immune responses are responsible for PD. The severity of this pathophysiological condition is dependent on several factors, including environmental and host risk factors such as smoking or socioeconomic status [7]. PD persists throughout periods of activity and quiescence until the microbial biofilm is therapeutically removed, or the affected tooth is extracted. The prevalence of periodontitis among dentate adults was 62% from 2011 to 2020, with severe periodontitis comprising 23.6% of cases [8]. Potential associations between PD and specific non-communicable chronic diseases have been widely proven. For instance, PD can be observed in autoimmune diseases like rheumatic arthritis and systemic lupus erythematosus, implicating periodontitis as a potential trigger of autoimmunity [9,10]. An exaggerated immune response in autoimmune disorders may lead to the worsening of PD.

It is intuitively possible to reason that the decreasing amount of saliva in the progression of SS harms the bacterial component of the periodontium. Raised antibodies to *Prevotella denticola* can be found in SS, compared to osteoarthritis or PD [11]. Dysbiosis of *Prevotella denticola* may further exacerbate PD. Additionally, in SS patients, the normal vascular structure of healthy gums is replaced by a cobweb configuration, presenting with greater vascular inflammation than in individuals without the condition [12,13]. In the opposite direction, aggravated periodontitis may result in retrograde salivary gland infections, with components such as bacterial endotoxins initiating pathogen-associated molecular patterns and thereby exacerbating the inflammatory state of SS [14]. A study revealed that the risk of developing SS is approximately 50% higher in individuals with PD [15].

Pathophysiologically, SS and PD are intricately linked, with specific pathways potentially mediated by complex networks of cells (e.g., γδT cells) and molecules (e.g., CXCR4) [16,17,18]. However, a clear association between SS and PD has not yet been vividly demonstrated (Figure 1). This study seeks to enhance the comprehension of the co-occurrence mechanisms of SS and PD by investigating their common transcriptional landscape. Through multi-source dataset integration and cross-disease target identification, this data-driven approach replaces hypothesis-driven research, furnishing a foundation for comorbidity prediction modeling and bioinformatics evidence for comorbidity subtype analysis.

## 2. Materials and Methods

### 2.1. Study Design and Methodology

The present study delineated the profiling of differentially expressed genes (DEGs) in individuals with SS and PD and conducted gene set enrichment analysis (GSEA). Weighted gene co-expression network analysis (WGCNA) was employed to identify genes from significant modules. The least absolute shrinkage and selection operator (LASSO) machine learning technique was employed to refine the DEGs, subsequent to the validation using the receiver operating characteristic (ROC) curve. Immune cell infiltration in SS and PD was conducted utilizing the cell-type identification by estimating relative subsets of RNA transcripts (CIBERSORT) approach (Figure 2).

### 2.2. Data Collection

Gene expression profiling data for SS (GSE51092) and PD (GSE16134) were sourced from the Gene Expression Omnibus (GEO, http://www.ncbi.nlm.nih.gov/geo) (accessed on 25 May 2025) and utilized as test sets to find shared genes between the two conditions. GSE40611 and GSE10334 served as external validation cohorts to verify the expression levels and diagnostic efficacy of the shared gene. Table 1 displays the attributes of the relevant microarray datasets.

### 2.3. Differentially Expressed Genes (DEGs) Analysis and Functional Enrichment Analysis

Co-variance batch adjustment (ComBat) and robust multi-array average (RMA) algorithms performed batch effect adjustment, background correction, and normalization before analysis. The R tool limma (version 3.64.1) was used to find DEGs meeting FDR < 0.05 and |Fold change| > 1.2 criteria. GSEA was used to evaluate DEGs distribution and phenotypic relationships. The annotation files from the Molecular Signatures Database (MSigDB) were evaluated using GSEA. Gene expression profiles and phenotypic groupings containing 5–5000 genes were employed. After 1000 resamplings, we confirmed statistical significance with a NES score >1, adjusted *p*-value < 0.05, and FDR < 0.25. Functional enrichment study used common DEGs with consistent expression patterns in the SS and PD test sets. We obtained the MSigDB subset c2.cp.kegg.v7.4.symbols.gmt for gene set functional enrichment analysis. The gene set enrichment data was obtained by mapping the genes to the background set and using clusterProfiler (version 4.16.0) in R to analyze it.

### 2.4. Weighted Gene Co-Expression Network Analysis (WGCNA)

The investigation commenced with the median absolute deviation (MAD) of each gene. Consequently, we eliminated 50% of genes exhibiting low MAD values. We constructed a scale-free co-expression network by filtering the expression matrix of DEGs with the goodSamplesGenes function. The pickSoftThreshold function ascertained gene adjacency. This function employs co-expression similarity to compute β, the soft-thresholding parameter. A topological overlap matrix (TOM) was generated from the adjacency matrices. Subsequently, we computed gene dissimilarity and ratio. Gene modules were identified by hierarchical clustering and dynamic tree-cutting techniques. We selected a truncation value for the module tree diagram. We incorporated modules according to their projected gene dissimilarity to illustrate the co-expression network of feature genes. WGCNA identified the most significant gene modules for SS and PD, resulting in the formation of shared DEGs.

### 2.5. Least Absolute Shrinkage and Selection Operator (LASSO) Machine Learning

The expression matrices of the shared DEGs in SS and PD after the intersection of DEGs analysis and WGCNA were used as characteristic values using the LASSO method. We used LASSO for binary classification diagnosis. First, we preprocessed the data, including removing genes with low expression and standardizing the features. We set the grouping information and applied LASSO regression for feature selection, choosing the optimal λ value through cross-validation. Then, we built a logistic regression model using the selected features. LASSO regression modeling reduces the risk of overfitting by constraining less significant coefficients towards zero, hence facilitating variable selection. The common characteristic gene can be obtained by taking the intersection of the Lasso analysis results of the test sets for SS and PD.

### 2.6. Evaluation of Expressions and Receiver Operating Characteristic (ROC) Curves

The R packages ggplot2 (version 3.5.2) and pROC (version 1.18.5) were utilized to conduct Student’s *t*-test comparing cases and controls and to generate ROC curves for evaluating the diagnostic accuracy of the common characteristic gene based on the area under the curve (AUC). An AUC greater than 0.7 was deemed indicative of strong discrimination.

### 2.7. Immune Infiltration Analysis

The CIBERSORT method profiled the abundance of 22 immune cell types, including B cells naïve, B cells memory, plasma cells, T cells CD8, T cells CD4 naïve, T cells CD4 memory resting, T cells CD4 memory activated, T cells follicular helper, T cells regulatory (Tregs), T cells gamma delta, NK cells resting, NK cells activated, monocytes, macrophages M0, macrophages M1, macrophages M2, dendritic cells resting, dendritic cells activated, mast cells resting, mast cells activated, eosinophils, and neutrophils. A correlation study between the shared gene and the commonly enriched immune cells was conducted utilizing Spearman’s approach.

### 2.8. Statistical Analysis

Statistical analyses were conducted with R software (version 4.3.1). DEGs were identified using the limma package with Benjamini–Hochberg correction, setting a threshold of FDR < 0.05. In GSEA, *p*-values were corrected via the Benjamini–Hochberg approach, with enriched gene sets deemed significant at FDR < 0.25 and nominal *p* < 0.05. In WGCNA, co-expression modules were identified with parameters optimized as follows: soft threshold power = 7/12 (R^2^ ≥ 0.8), minModuleSize = 30, deepSplit = 3, and mergeCutHeight = 0.25. Module membership (MM) > 0.8 and gene significance (GS) > 0.1 discovered hub genes. To ascertain the regularization parameter λ in LASSO regression, 10-fold cross-validation was employed, utilizing the 1-SE rule to reconcile model parsimony with predictive accuracy. An independent sample’s *t*-test was employed to assess *CCR1* expression following the confirmation of normality by the Shapiro–Wilk test (*p* > 0.05). The ROC curve analysis utilized 95% confidence intervals to assess the area under the curve via the DeLong method. Analysis of immune cell infiltration employed Spearman’s rank correlation to assess the relationship between *CCR1* and immune cell subsets, with statistical significance defined as *p* < 0.05 (i.e., −log10 (*p*-value) > 1.3).

## 3. Results

### 3.1. Identification of DEGs and GSEA Pathways of SS and PD in the Test Sets

The GSE51092 dataset found 835 DEGs, 448 of which were up-regulated and 387 down-regulated (Figure 3A). The GSE16134 dataset found 3608 DEGs, including 1795 up-regulated and 1813 down-regulated genes (Figure 3B). GSE51092 DEGs were concentrated in RNA polymerase, NOD-like receptor signaling pathway, and RIG-I-like receptor signaling pathway (Figure 3C). DEGs in GSE16134 were enriched in glycosphingolipid biosynthesis ganglio series, antigen processing and presentation, and systemic lupus erythematosus (Figure 3D). After taking the intersection of the test sets of SS and PD, 66 up-regulated genes and 29 down-regulated genes were obtained, totaling 95 common DEGs with consistent expression trends (Figure 3E). The 95 common DEGs were mainly enriched in five terms, cytokine–cytokine receptor interaction, chemokine signaling pathway, proteasome, intestinal immune network for IgA production, and cytosolic DNA sensing pathway (Figure 3F).

### 3.2. WGCNA of SS and PD in the Test Sets

Figure 4A shows that the optimal soft-thresholding value β in the GSE51092 dataset is seven. Merge modules used a 0.25 dynamic tree cutoff. GSE51092 had 26 modules after similarity-based aggregation (Figure 4C). The Pearson correlation coefficients of each module with sample attributes were calculated. The brown module had a positive correlation with SS (*p* = 8.5 × 10^−9^, r = 0.37) (Figure 4E). In the brown module, module membership exhibited a positive correlation with SS gene significance (*p* = 6.7 × 10^−49^, r = 0.69) (Figure 4G). Figure 4B shows that the appropriate soft-thresholding value β for the GSE16134 dataset is 12. Similarity-based clustering and a dynamic tree cutoff value of 0.25 for module amalgamation identified 19 GSE16134 modules (Figure 4D). The Pearson correlation coefficients for each module with sample attributes were computed. The blue module had a significant positive correlation with PD (*p* = 8.5 × 10^−38^, r = 0.64) (Figure 4F). In the blue module, membership exhibited a positive correlation with gene significance for PD (*p* = 5.7 × 10^−213^, r = 0.77) (Figure 4H).

### 3.3. Interacted DEG of SS and PD Using LASSO Regression

Upon intersecting the brown module in the SS test set with the blue module in the PD test set after WGCNA, 39 genes were obtained (Figure 5A). After intersecting DEGs analysis results with those of WGCNA in the SS and PD test sets, 16 genes were derived as the input data for Lasso regression (Figure 5B). To gain deeper insights into the characteristic genes of both SS and PD, LASSO regression analysis was performed using the matrices of 16 intersecting DEGs (Figure 5C–F). Through the screening process, three intersecting DEGs were detected in SS and nine in PD. Notably, *CCR1* was found to be a candidate pathogenic biomarker between the two conditions (Figure 5G).

### 3.4. Expression Levels and ROC Curves of CCR1 in SS and PD

The expression levels of *CCR1* in SS and PD, illustrated in violin plots, exceeded those in the control groups (Figure 6A,C,E,G). ROC curves were constructed to assess the diagnostic efficacy of the *CCR1* gene. Within the framework of SS, *CCR1* displayed commendable diagnostic efficacy in distinguishing cases from controls, achieving an AUC of 0.74 in the test set and 0.83 in the validation set (Figure 6B,F). In PD, *CCR1* had superior diagnostic performance in differentiating cases from controls, achieving an AUC of 0.83 in the test set and 0.81 in the validation set (Figure 6D,H).

### 3.5. Immune Cell Infiltration in SS and PD

Significant disparities in immune cell profiles were seen between the illness groups and the control group (Figure 7A,E). In the salivary glands of SS individuals, the levels of B cells naïve, T cells CD4 naïve, T cells CD4 memory activated, and T cells gamma delta were elevated, while the levels of plasma cells, T cells CD8, NK cells resting, monocytes, and mast cells activated were degraded (Figure 7B). Compared with healthy controls, PD gingival tissues showed higher levels of plasma cells, T cells CD4 naïve, T cells CD4 memory activated, T cells gamma delta, dendritic cells resting, and neutrophils. Conversely, the levels of B cells memory, T cells CD8, T cells follicular helper, T cells regulatory (Tregs), macrophages M1, macrophages M2, dendritic cells activated, and mast cells were lower (Figure 7F). Notably, in both SS and PD tissues, the levels of T cells CD4 naïve, T cells CD4 memory activated, and T cells gamma delta were increased, and the levels of T cells CD8 were decreased, showing a similar trend. The heatmaps illustrating correlations among diverse immune cells revealed that the illness groups exhibited a unique immunological profile in contrast to the control group, while also emphasizing the relationships between different immune cell types (Figure 7C,G). Furthermore, we examined the correlations between the *CCR1* gene and immune cells in patients with SS and PD. In SS, B cells naïve, T cells CD4 naïve, T cells gamma delta, and macrophages M2 were significantly positively correlated with *CCR1*, while plasma cells, NK cells resting, NK cells activated, and monocytes showed an inverse correlation. In PD, plasma cells, T cells CD4 naïve, T cells CD4 memory activated, T cells gamma delta, macrophage M0, and neutrophils were significantly positively correlated with *CCR1*, while T cells CD8, T cells follicular helper, T cells regulatory (Treg), NK cells activated, monocytes, macrophage M1, macrophage M2, dendritic cells resting, dendritic cells activated, and mast cells resting also showed an inverse correlation (Figure 7D,H). In both SS and PD, T cells CD4 naïve and T cells gamma delta exhibit a positive correlation with *CCR1*, whereas NK cells activated and monocytes show a negative correlation with *CCR1*. The data indicate that *CCR1* may be pivotal in the development of SS and PD by affecting immune cell infiltration.

## 4. Discussion

Fairly, evidence suggests that SS is linked to a heightened risk of caries. Evidence suggests that SS and PD may serve as mutual hazards; nevertheless, this assertion remains contentious. A meta-analysis of nine studies (encompassing 198 SS cases and 180 controls), a meta-analysis of 10 cross-sectional studies (encompassing 228 SS cases and 223 controls), and a systematic review of 17 studies (encompassing 518 SS cases and 544 controls) concluded that, despite the observation of elevated periodontal burden indices, such as plaque index and gingival index, the collective evidence provided by these studies was insufficient to suggest a SS and PD causality [19,20,21]. However, compared to controls, patients with SS are prone to utilize ambulatory dental services for PD at a significantly higher rate in the three years before diagnosis [22]. A recent genetic instrumental variable analysis, including a substantial sample size (comprising 2247 cases of SS, 332,115 controls of SS, 17,353 cases of PD, and 28,210 controls of PD), indicates that SS facilitates PD [23]. Retrospective cohort research involving 76,765 cases and controls of PD suggests that the incidence of sepsis was significantly heightened in patients with PD compared to the general population [24]. The association between PD and SS was especially significant in the female group. The use of ambulatory dentistry services for PD is markedly higher in individuals with SS over the three years preceding diagnosis compared to the control group [25]. The presence of SS appears to adversely impact periodontal health, as gingival inflammation is more pronounced in patients with this condition. Salivary BAFF may partially mediate the established influence of B cells in SS patients [26]. Individuals with SS exhibit a diminished salivary flow rate, along with an increased periodontal index and an imbalance of pro- and anti-inflammatory cytokines. The management of PD in SS patients has been shown to yield increased salivary flow and improved ESSPRI [27]. PD has been observed to alter the microRNA profile involved in the salivary glands’ inflammatory process [14]. However, the precise mechanism by which this leads to lesions in the salivary glands of patients with SS remains to be elucidated.

In our study, in the SS dataset, DEG-enriched pathways were mainly linked to RNA polymerase, which is crucial for gene transcription and subsequent protein synthesis and cell functions [28]. NOD-like receptors are essential cytoplasmic pattern-recognition receptors involved in the host innate immune response, immunological homeostasis, and autoimmune diseases [29]. The RIG-I-like receptor signaling pathway can recognize viral infections and initiate immune responses; its abnormal activation or defects are related to autoimmune diseases like systemic lupus erythematosus [30]. In the PD dataset, DEG-enriched pathways were mainly associated with glycosphingolipid biosynthesis ganglio series, which alters autophagy in osteoclasts [31]. Antigen processing and presentation, which is a key immune process for antigen recognition, where antigen-presenting cells take in antigens and show MHC-peptide complexes to T cells for immune response initiation [32]. Systemic lupus erythematosus, where both T cells with abnormal cytokine secretion and regulation and B cells with over-production of autoantibodies and antigen presentation cause immune tolerance breakdown and tissue damage, similar to PD [33].

*CCR1* has been reported to be associated with periodontitis using multiple computational tools [34]. The lack of chemokine receptor *CCR1*^+^ diminishes inflammatory bone resorption throughout experimental periodontitis in mice. Inhibition of *CCR1* and suppression of osteoclast differentiation can be observed in murine experimental periodontitis [35]. Inhibition of the *CCR1* gene in macrophages of periodontitis may reduce the chemotaxis of local macrophages in the periodontal region, which may contribute to the alleviation of bone destruction caused by periodontitis [36]. Bioinformatics analysis has integrated three salivary gland datasets of SS and screened *CCR1* as a hub gene. Immunohistochemical investigations have revealed that the expression of *CCR1* protein in the labial glands of SS patients is elevated compared to non-SS patients [37]. CFL1 can modulate the expression levels of the downstream CCL5/*CCR1* axis to augment the migration and proliferation of bone marrow-derived mesenchymal stem cells in SS murine models [38]. RNA sequencing revealed a set of autosomal differentially expressed genes with an interferon signature in the B cells of SS, notably including highly elevated *CCR1*, which has been verified [39]. *CCR1* is implicated in various inflammatory illnesses and is linked to particular immune cells. The CCL2/CCR2 axis and CCL3 can recruit circulating monocytes to the inflamed synovium through *CCR1* [40]. Eosinophil-derived hCCL15/23 and mCCL6 can engage with *CCR1* to facilitate eosinophilic airway inflammation [41]. The ligands of *CCR1* impede the degranulation and mediator release of mast cells activated by IgE/Ag, SCF, or HSP70 via the phosphorylation of Lck and the dephosphorylation of the phosphatase-1 containing the Src homology 2 domain [42]. *CCR1* antagonists can impede cell infiltration, inflammation, and the Th1 cytokine response in human *CCR1* transgenic mice [43].

An examination of immune cell infiltration disclosed the features of immune cells within the salivary glands of SS. In comparison to the salivary glands of healthy individuals, the levels of γδ T cells and naïve CD4^+^ T cells were dramatically elevated, although the quantity of plasma cells diminished markedly [44]. In individuals with SS, the number of γδ T cells and naïve CD4^+^ T cells in peripheral blood is decreased compared to healthy controls. The increasing number of γδ T and naïve CD4^+^ T cells in the salivary glands indicates that γδ T cells may be chemotactically drawn to the inflammatory regions of the salivary glands [16,45]. Peripheral lymphopenia in SS is linked to the early senescence of naïve CD4^+^ T cells [16]. Natural killer cells regulate CD8^+^ T cells in the submandibular gland of SS through the PD-1-PD-L1 pathway [46]. Monocytes/macrophages are linked to the facilitation of inflammation, overactivation of lymphocytes, and subsequent structural damage in SS [47]. In SS-associated sialadenitis, mast cells are situated next to fibroblasts and secrete TGFβ1, which stimulates collagen production in fibroblasts and promotes fibrosis [48]. γδ T cells constitute the predominant T cell population at the epithelial barrier and are regarded as regulators of local immunity in the gingiva, facilitating the maintenance of host-biofilm balance. γδ T lymphocytes contribute negatively to oral infection models by facilitating the pathogen-induced, bone-destructive immune response in periodontal disease [17]. Through the secretion of cytokines and the activation of osteoclasts, CD4^+^ T cells are responsible for the inflammation associated with periodontal disease. The principal helper T cell subsets are Th1/Th2 and Th17/Treg, which develop from naïve CD4^+^ T cells [49]. NK cells promote pro-inflammatory responses via cytotoxic reactions, chemokine/cytokine production, B/T cell regulation, autoimmunity upregulation, and dendritic cell crosstalk. They can also modulate B, T, and dendritic cells by downregulating autoimmunity for oral health restoration in PD [50]. In PD-affected tissues, there was a significantly higher proportion of intermediate (CD14CD16) monocytes than in healthy tissues. These monocytes overexpress HLA-DR and PDL1, showing their activated inflammatory state [51]. Mast cells may be linked to collagen maturation in periodontal tissues during the initial phase of periodontal disease etiology [52].

Several studies have examined the genetic basis of SS and PD. However, few have employed bioinformatics to investigate the potential links between the two conditions. This work employed bioinformatics to find a pathogenic potential biomarker linked to SS and PD. This method enabled a more profound comprehension of the underlying mechanisms that lead to the emergence of both situations. Nevertheless, it is important to acknowledge the limitations of our study. First, the lack of in vitro/in vivo experimental validation (e.g., cell-based assays, animal models) precludes direct verification of bioinformatics-predicted molecular mechanisms, gene regulatory networks, and pathway functions. Consequently, findings remain correlational, impeding causal inference and mechanistic insights. Second, reliance on public database data introduces risks of sample heterogeneity and bias, potentially leading to overfitting or false positives in the absence of experimental validation. Third, the absence of preclinical validation limits the direct applicability of bioinformatics findings to clinical practice, confining conclusions to theoretical frameworks. Fourth, missing data on patients with comorbid SS and PD hinders the assessment of synergistic pathogenic mechanisms and immune microenvironment interactions. This gap undermines understanding of disease complexity in real-world settings. Notwithstanding these constraints, the preliminary identification of the biomarker may establish a foundation for novel possibilities in subsequent research.

*CCR1* holds potential as an early diagnostic biomarker and longitudinal predictor of disease progression/relapse in SS and PD. *CCR1* antagonists (e.g., BX471) may block CCR1-mediated chemotaxis of immune cells (e.g., monocytes, T cells) to inflamed tissues in SS and PD [53,54]. However, *CCR1* shares homology with other chemokine receptors (e.g., CCR5), risking off-target effects. Developing highly selective antagonists remains challenging. Systemic *CCR1* blockade may impair normal immune surveillance, increasing susceptibility to infections or malignancies. Biomarker-guided patient stratification (e.g., *CCR1*-high subgroups) could mitigate these risks. Future studies will perform the following: (1) integrate genomic/proteomic/epigenetic data to refine patient subsets and combinatorial targets; (2) validate differential genes in an SS-PD comorbidity model (NOD/Ltj mice with periodontal ligation) via therapeutic intervention and mechanistic assays; and (3) conduct a biomarker-guided clinical trial in SS-PD patients, integrating subgroup analysis of sensitive genes with *CCR1*-targeted therapies, following bioinformatics validation via cellular/molecular assays.

## 5. Conclusions

In both SS and PD, *CCR1* is overexpressed. T cells CD4 naïve and T cells gamma delta show a positive correlation with *CCR1* expression in the context of these diseases, while NK cells activated and monocytes exhibit a negative correlation with *CCR1* expression. *CCR1* serves as a critical pathogenic hub linking SS and PD (Figure 8). *CCR1* will become a robust biomarker for the concurrent diagnosis of SS and PD, which may yield innovative insights for forthcoming investigations into the pathogenesis, diagnosis, and treatment of SS and PD. This study lacks in vitro/in vivo validation and data from SS-PD comorbid patients. Future work will obtain ex vivo tissues from such patients to validate *CCR1* expression and explore its mechanism linking SS and PD via molecular and cellular biology assays.

## Figures and Tables

**Figure 1 cimb-47-00523-f001:**
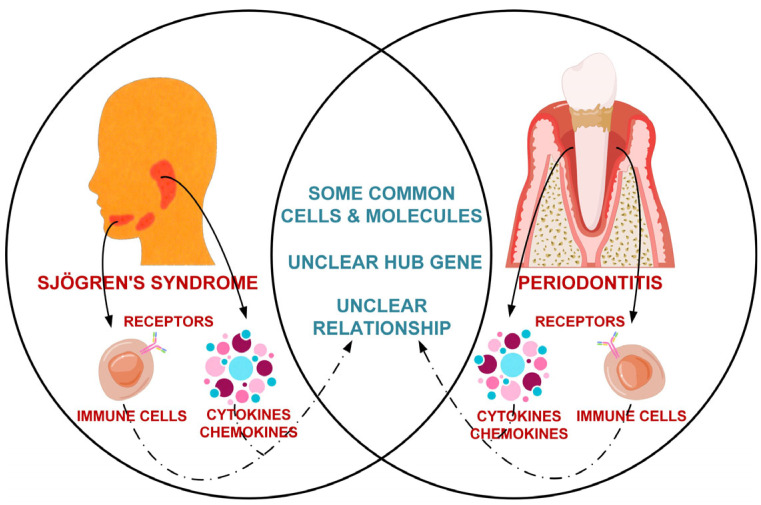
The physiopathological background of Sjögren’s syndrome (SS) and periodontitis (PD).

**Figure 2 cimb-47-00523-f002:**
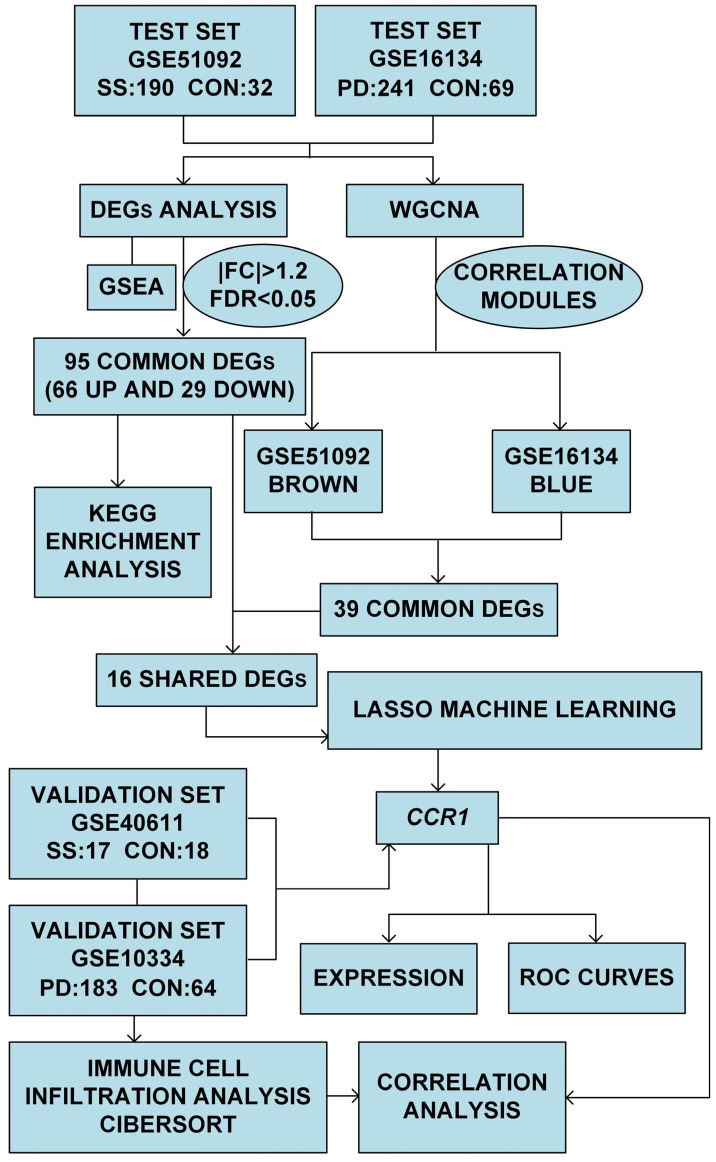
The framework for data analysis.

**Figure 3 cimb-47-00523-f003:**
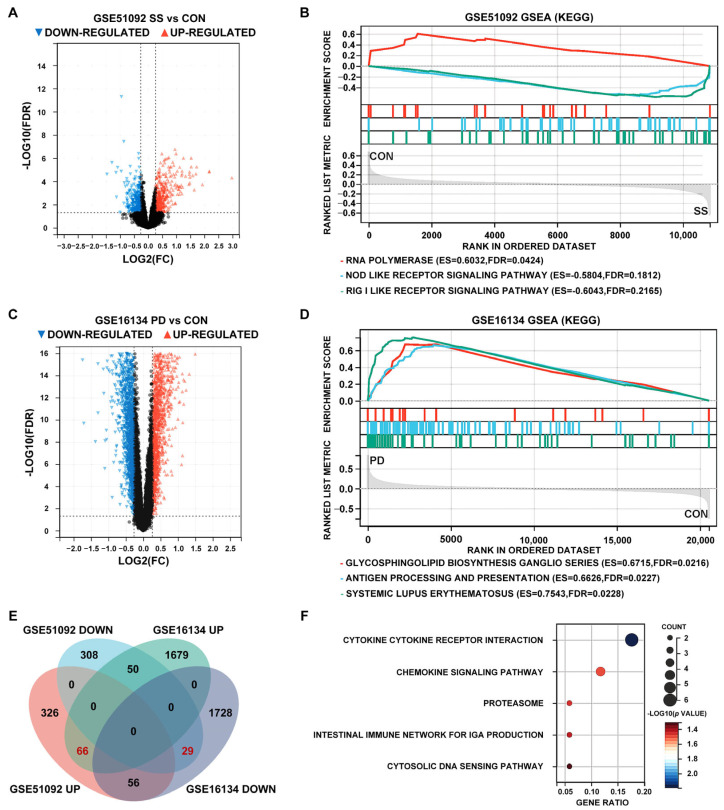
Identification of differentially expressed genes (DEGs) and gene set enrichment analysis (GSEA) pathways of SS and PD. (**A**,**C**) Volcano maps of SS and PD test sets. (**B**,**D**) KEGG enrichment analysis of SS and PD test sets using the GSEA method. (**E**) Intersection of DEGs in SS and PD test sets. (**F**) Kyoto encyclopedia of genes and genomes (KEGG) enrichment analysis of common DEGs.

**Figure 4 cimb-47-00523-f004:**
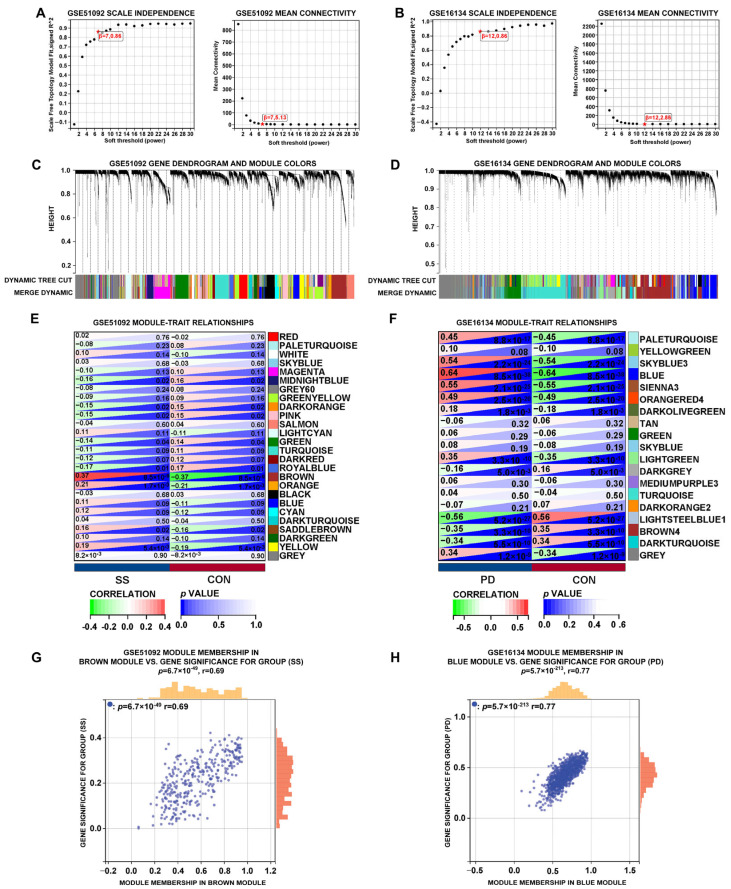
Weighted gene co-expression network analysis (WGCNA) of the significant modules in SS and PD. (**A**,**B**) The selection of soft thresholds in SS (GSE51092) and PD (GSE16134). (**C**,**D**) Gene cluster dendrograms for SS and PD by the dynamic tree cut algorithm. (**E**,**F**) Heatmaps of SS and PD module-clinical trait associations. (**G**,**H**) The relationships between module membership and gene significance in the brown module of SS and the blue module of PD.

**Figure 5 cimb-47-00523-f005:**
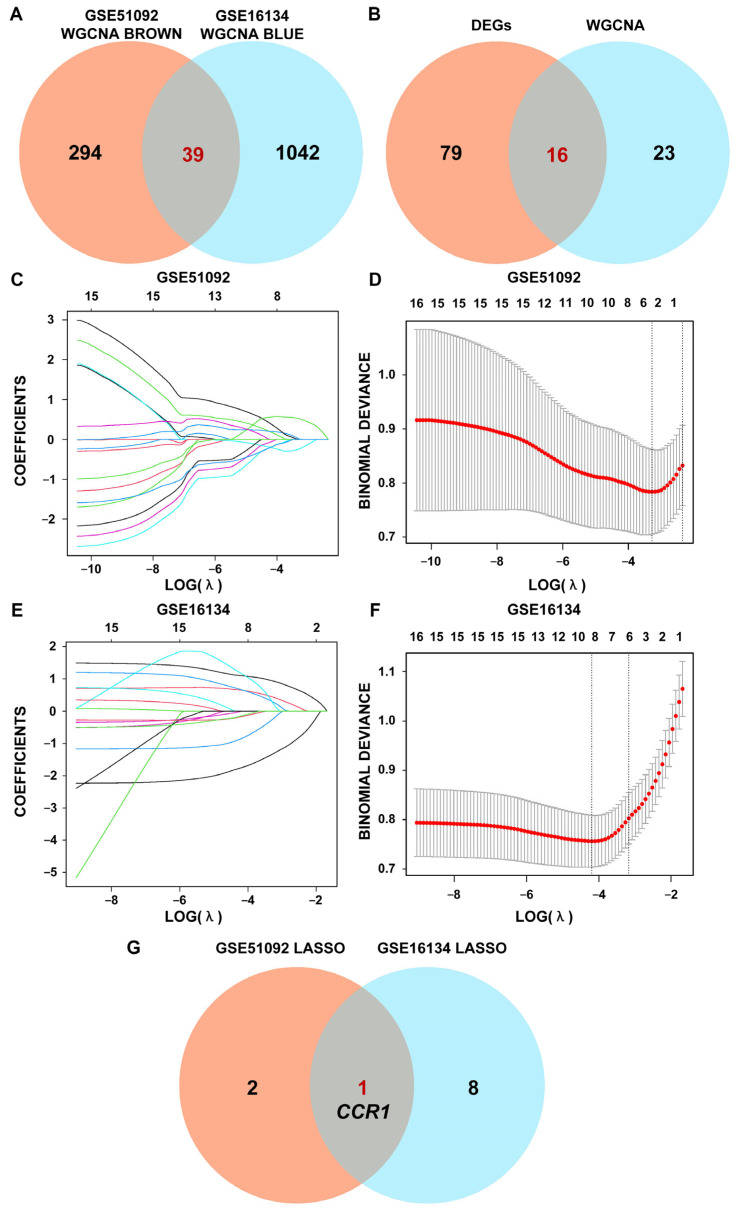
Interacted DEGs of SS and PD using least absolute shrinkage and selection operator (LASSO) regression. (**A**) Intersection of the brown module of SS and the blue module of PD. (**B**) Intersection of DEGs analysis and WGCNA. (**C**,**E**) The change curve of characteristic genes for SS and PD. (**D**,**F**) The outcome of cross-referencing lambda results in the SS and PD test sets. (**G**) The overlapped genes of LASSO regression analysis in the SS and PD test sets by the Venn diagram.

**Figure 6 cimb-47-00523-f006:**
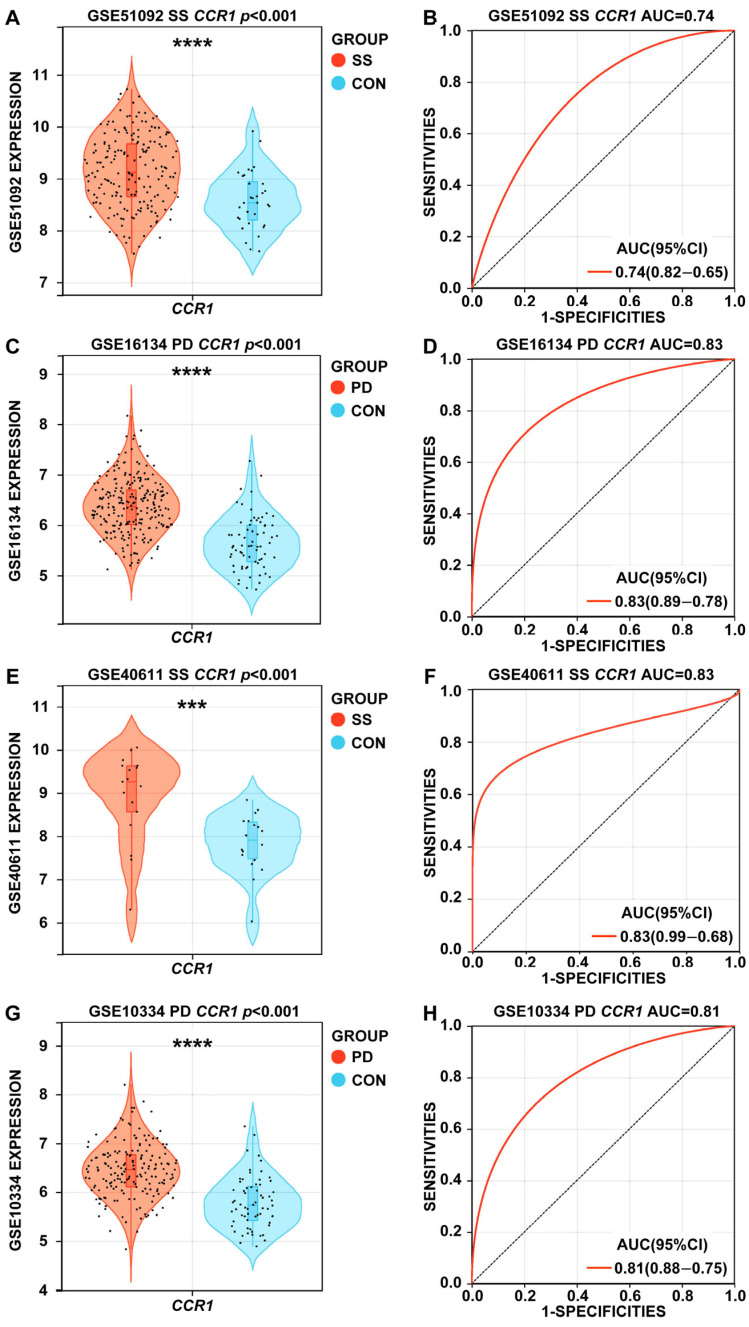
Expression levels and receiver operating characteristic (ROC) curves of *CCR1* in SS and PD. (**A**,**E**) The expression values of *CCR1* in the SS test and validation set. (**B**,**F**) The ROC curve of *CCR1* in the SS test and validation set. (**C**,**G**) The expression values of *CCR1* in the PD test and validation set. (**D**,**H**) The ROC curve of *CCR1* in the PD test and validation set. *** *p* < 0.001; **** *p* < 0.0001.

**Figure 7 cimb-47-00523-f007:**
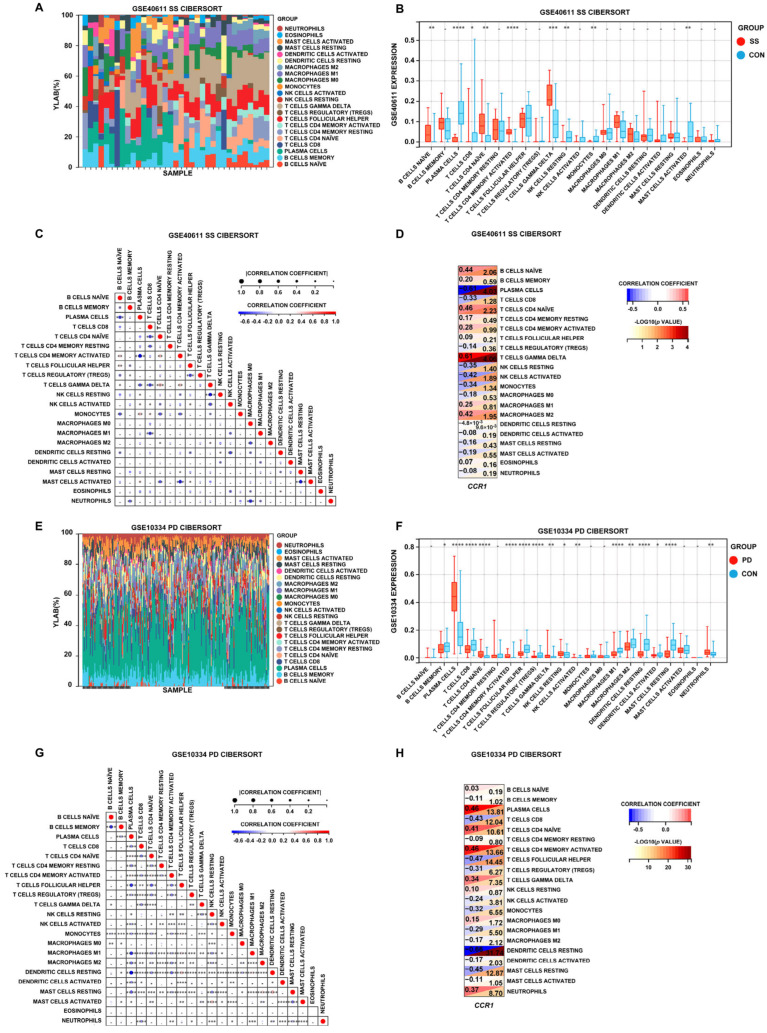
Immune cell infiltration in SS and PD. (**A**,**E**) Stacked bar charts depicting the proportions of immune cell infiltration in the tissues of SS and PD patients. (**B**,**F**) Box plots comparing the proportions of immune cells between the SS and PD groups and the control group. (**C**,**G**) Heatmaps illustrating the correlations among different immune cells in the tissues of SS and PD patients. The asterisks within the squares denote the correlation coefficients between the respective immune cells. (**D**,**H**) Correlations between *CCR1* and immune cells in SS and PD. * *p* < 0.05; ** *p* < 0.01; *** *p* < 0.001; **** *p* < 0.0001.

**Figure 8 cimb-47-00523-f008:**
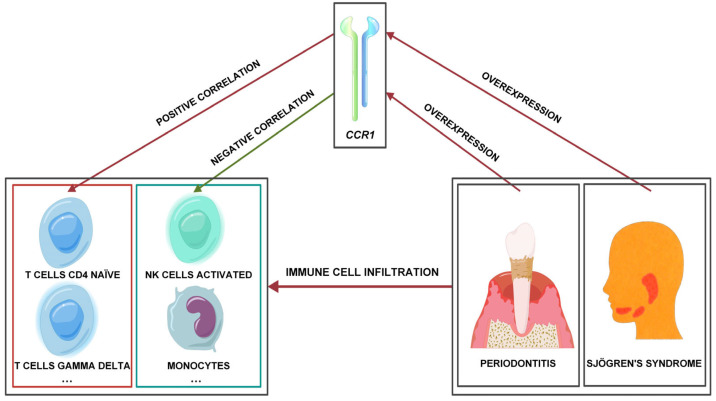
*CCR1* serves as a pathogenic hub linking SS and PD.

**Table 1 cimb-47-00523-t001:** Characteristics of the involved microarray datasets.

Accession	Type	Disease	Control	Case	Source	Platform
GSE51092	Test	SS	32	190	Blood	GPL6884
GSE16134	Test	PD	69	241	Gingiva	GPL570
GSE40611	Validation	SS	18	17	Parotid	GPL10558
GSE10334	Validation	PD	64	183	Gingiva	GPL570

## Data Availability

The data presented in this study are available in the GEO database at https://www.ncbi.nlm.nih.gov/geo/. These data were derived from the following resources available in the public domain: https://www.ncbi.nlm.nih.gov/geo/query/acc.cgi?acc=GSE51092 (accessed on 25 May 2025). https://www.ncbi.nlm.nih.gov/geo/query/acc.cgi?acc=GSE16134 (accessed on 25 May 2025). https://www.ncbi.nlm.nih.gov/geo/query/acc.cgi?acc=GSE40611 (accessed on 25 May 2025). https://www.ncbi.nlm.nih.gov/geo/query/acc.cgi?acc=GSE10334 (accessed on 25 May 2025).

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
