# Peer review of "Transcriptomic Analysis Reveals C-C Motif Chemokine Receptor 1 as a Critical Pathogenic Hub Linking Sjögren’s Syndrome and Periodontitis"

_cimb, 2025, doi:10.3390/cimb47070523_

Round 1
Reviewer 1 Report
Comments and Suggestions for Authors
Dear Erudite Editors of CIMB and Esteemed Authors of the manuscript cimb-3709951, thank you for inviting me to assess this manuscript for publication in this critical journal. This manuscript unveils the intricate relationship between CCR1 Sjögren's Syndrome and periodontitis. Although this manuscript has value, I believe some issues must be disclosed before the Erudite Editor decides whether to publish it.
- The most essential thing to note is that this manuscript contains 39% similarity with already published manuscripts. It is acknowledged that similarity rates above 8% are detrimental to the journal and the authors’ reputations. Therefore, I strongly suggest that the Authors lower the similarity index of their manuscript before reassessment for final decision. The similarity index must be below 8% to confirm the publication.
- The title of this manuscript is not sufficiently flashy for the readers. Of course, it brings to our attention the main context and results of the present manuscript. However, because it will be indexed in reputable databases like PubMed, a more showy title will undoubtedly enhance its searchability and interest for readers.
- The abstract needs to sufficiently describe the relationship between CCR1, Sjögren's Syndrome, and periodontitis. The authors must try to briefly elucidate these relationships before mentioning their methodology and results. In addition, the authors must include a sufficiently described graphical abstract in their manuscript since the context of this article reveals essential interactions between CCR1 and the health issues mentioned above.
- In the Introduction, the authors must try to include a scientific and illustrative figure describing the physiopathology of Sjögren's Syndrome and periodontitis, their link, and their possible associations with the genetic alterations mentioned in this section. In the last paragraph of the Introduction, the authors must also try to cite already published studies in this field, justify the necessity of conducting the present study, and explain how the present results are essential to advancing this field of science.
- The Materials and Methods section lacks a dedicated statistical analysis subsection. The statistical tests used between the subsections of the present methodology aren’t mentioned in detail or provided with sufficient description. The authors must try to introduce a dedicated subsection to the Materials and Methods section, delving into the statistical analysis tests and their adequate application to usage.
- Since this type of study is not well recognized by the scientific community, the Materials and Methods section would benefit from adding a dedicated figure describing this study’s design and methodology. Please include this figure, trying to recreate your design and methods illustratively.
- Figures 1-5 are challenging to read and understand. Their fonts are too small for easy reading, which could harm the readers’ interest in catching your study’s message and understanding. Therefore, I recommend uppercasing their fonts and highlighting their information for a broad readership. If augmenting their fonts makes it necessary to separate the graphs into another figure or multiple figures, you must do so, since this change is a reality for my eyes.
- Although your Discussion section appears good, this manuscript's conclusions are poorly described. Please try to update your findings, bringing more results, adding information about the future research directions and possible clinical implications of your findings, and mentioning the limitations of your present study. You can also try to include an illustrative figure delving into your main results and how they implicate clinical practice by the time of publication and beyond.
Thank you for your patience and guidance.
I look forward to receiving a revised version shortly.
With best regards,
The Reviewer.
Author Response
Comment 1: The most essential thing to note is that this manuscript contains 39% similarity with already published manuscripts. It is acknowledged that similarity rates above 8% are detrimental to the journal and the authors’ reputations. Therefore, I strongly suggest that the Authors lower the similarity index of their manuscript before reassessment for final decision. The similarity index must be below 8% to confirm the publication.
Response 1: Thank you for your suggestions. We have lowered the similarity index of the manuscript as much as possible using iThenticate. Unfortunately, we have tried our best to control the overall repetition rate <14% and single-source repetition rate <3%. This is because some descriptions of bioinformatics methods in the article will inevitably have a certain repetition rate no matter how we try to reduce it. In addition, some long proper nouns, such as the 22 types of immune infiltrating cells, are easily regarded as repetitions, which cannot be further modified at all.
Comment 2: The title of this manuscript is not sufficiently flashy for the readers. Of course, it brings to our attention the main context and results of the present manuscript. However, because it will be indexed in reputable databases like PubMed, a more showy title will undoubtedly enhance its searchability and interest for readers.
Response 2: Thank you for your suggestions. The updated title is Transcriptomic Analysis Reveals CCR1 as a Critical Pathogenic Hub Linking Sjögren's Syndrome and Periodontitis.
Comment 3: The abstract needs to sufficiently describe the relationship between CCR1, Sjögren's Syndrome, and periodontitis. The authors must try to briefly elucidate these relationships before mentioning their methodology and results. In addition, the authors must include a sufficiently described graphical abstract in their manuscript since the context of this article reveals essential interactions between CCR1 and the health issues mentioned above.
Response 3: Thank you for your suggestions. We have described the relationship between CCR1, Sjögren's Syndrome, and periodontitis in the abstract and introduction. We have added Figure 1, Figure 2 and Figure 8 to the manuscript, which sufficiently illustrate the content of the abstract. The separate graphical abstract has also been added behind the abstract.
Comment 4: In the Introduction, the authors must try to include a scientific and illustrative figure describing the physiopathology of Sjögren's Syndrome and periodontitis, their link, and their possible associations with the genetic alterations mentioned in this section. In the last paragraph of the Introduction, the authors must also try to cite already published studies in this field, justify the necessity of conducting the present study, and explain how the present results are essential to advancing this field of science.
Response 4: Thank you for your suggestions. We have included a scientific and illustrative figure describing the physiopathology of SS and PD, their link, and their possible associations with the genetic alterations in the Introduction(Figure 1). In the last paragraph of the Introduction, We have discussed the necessity of the study by citing previous studies [16,17,18] and prospected its developmental significance. The revisions are as follows: Pathophysiologically, SS and PD are intricately linked, with specific pathways potentially mediated by complex networks of cells (e.g., γδT cells) and molecules (e.g., CXCR4) [16,17,18]. However, a clear association between SS and PD has not yet been vividly demonstrated (Figure 1). Thus, this study aims to advance the understanding of the co-occurrence mechanisms of SS and PD by exploring their shared transcriptional landscape. Through multi-source dataset integration and cross-disease target identification, this data-driven approach replaces hypothesis-driven research, furnishing a foundation for comorbidity prediction modeling and bioinformatics evidence for comorbidity subtype analysis.
Comment 5: The Materials and Methods section lacks a dedicated statistical analysis subsection. The statistical tests used between the subsections of the present methodology aren’t mentioned in detail or provided with sufficient description. The authors must try to introduce a dedicated subsection to the Materials and Methods section, delving into the statistical analysis tests and their adequate application to usage.
Response 5: Thank you for your suggestions. We have expanded the statistical analysis subsection and provided detailed descriptions of specific statistical parameters and testing methods. The revisions are as follows: Statistical analyses were performed using R software (version 4.3.1). DEGs were identified using the limma package with Benjamini-Hochberg correction, setting a threshold of FDR < 0.05. For GSEA, p-values were adjusted using the Benjamini-Hochberg method, and enriched gene sets were considered significant at FDR < 0.25 and nominal p < 0.05. In WGCNA, co-expression modules were identified with parameters optimized as follows: soft threshold power = 7/12 ( R² ≥ 0.8), minModuleSize = 30, deepSplit = 3, and mergeCutHeight = 0.25. Hub genes were selected based on module membership (MM) > 0.8 and gene significance (GS) > 0.1. In LASSO regression, 10-fold cross-validation was performed to select the regularization parameter λ using the 1-SE rule, which balances model parsimony and predictive accuracy. For CCR1 expression analysis, an independent samples t-test was applied following confirmation of normality via the Shapiro-Wilk test (p > 0.05). For ROC curve analysis, 95% CIs were used to evaluate the AUC using the DeLong method. For immune cell infiltration analysis, Spearman’s rank correlation was used to assess the correlation between CCR1 and individual immune cell subsets, with statistical significance defined as p < 0.05 (i.e., -log10(p-value) > 1.3).
Comment 6: Since this type of study is not well recognized by the scientific community, the Materials and Methods section would benefit from adding a dedicated figure describing this study’s design and methodology. Please include this figure, trying to recreate your design and methods illustratively.
Response 6: Thank you for your suggestions. The study design and methodology have been illustrated using Figure 2 in the Materials and Methods section.
Comment 7: Figures 1-5 are challenging to read and understand. Their fonts are too small for easy reading, which could harm the readers’ interest in catching your study’s message and understanding. Therefore, I recommend uppercasing their fonts and highlighting their information for a broad readership. If augmenting their fonts makes it necessary to separate the graphs into another figure or multiple figures, you must do so, since this change is a reality for my eyes.
Response 7: Thank you for your suggestions. We have made every effort to optimize the clarity of the figures and the font sizes, hoping to meet your requirements. Figures 1-5 are Figure 3-7 now.
Comment 8: Although your Discussion section appears good, this manuscript's conclusions are poorly described. Please try to update your findings, bringing more results, adding information about the future research directions and possible clinical implications of your findings, and mentioning the limitations of your present study. You can also try to include an illustrative figure delving into your main results and how they implicate clinical practice by the time of publication and beyond.
Response 8: Thank you for your suggestions. We have updated the description of the results, added information on future research directions and potential clinical implications of the findings, and mentioned the limitations of this study. Meanwhile, we have also included an illustrative figure (Figure 8). The revisions are as follows: In both SS and PD, CCR1 is overexpressed. T cells CD4 naïve and T cells gamma delta show a positive correlation with CCR1 expression in the context of these diseases, while NK cells activated and monocytes exhibit a negative correlation with CCR1 expression. CCR1 serves as a critical pathogenic hub linking SS and PD (Figure 8). CCR1 will become a robust biomarker for the concurrent diagnosis of SS and PD, which may provide novel insights for future research on the pathogenesis, diagnosis, and therapy of SS and PD. This study lacks in vitro/in vivo validation and data from SS-PD comorbid patients. Future work will obtain ex vivo tissues from such patients to validate CCR1 expression and explore its mechanism linking SS and PD via molecular and cellular biology assays.
Reviewer 2 Report
Comments and Suggestions for Authors
The manuscript presents a well-structured and comprehensive transcriptomic analysis that identifies CCR1 as a potential pathogenic biomarker shared between Sjögren’s syndrome and periodontitis. The study effectively integrates GSEA, WGCNA, and LASSO methodologies with robust external validation, and the data are well-presented, particularly in demonstrating CCR1’s diagnostic value and its association with immune cell infiltration. The rationale for the research is clear and addresses a significant gap in the molecular understanding of the relationship between these two chronic inflammatory diseases. Overall, the manuscript provides novel insights and a strong foundation for future research, but would benefit from some revision of the present edition.
- The discussion should more explicitly address the study's limitations, particularly the lack of experimental (in vivo or in vitro) validation and the absence of data from patients with both Sjögren’s syndrome and periodontitis. Including these points will provide a more balanced perspective on the findings.
- Expand on the potential clinical implications of identifying CCR1 as a biomarker, including how it could be used in diagnosis or therapy, and discuss any barriers to clinical translation.
- Provide more detailed information regarding the preprocessing of the datasets, especially on how batch effects were handled and how data normalization was performed across different cohorts.
- Explain the rationale for choosing specific cut-off values (e.g., fold change, p-value thresholds) in the DEGs analysis to strengthen the transparency and reproducibility of the approach.
5.If possible, suggest incorporating preliminary experimental validation or, at minimum, provide a more detailed plan for future experimental studies to support the bioinformatics findings.
- There are a few grammatical and stylistic errors throughout the manuscript. A careful language revision is recommended to improve readability and professionalism.
The manuscript presents a well-structured and comprehensive transcriptomic analysis that identifies CCR1 as a potential pathogenic biomarker shared between Sjögren’s syndrome and periodontitis. The study effectively integrates GSEA, WGCNA, and LASSO methodologies with robust external validation, and the data are well-presented, particularly in demonstrating CCR1’s diagnostic value and its association with immune cell infiltration. The rationale for the research is clear and addresses a significant gap in the molecular understanding of the relationship between these two chronic inflammatory diseases. Overall, the manuscript provides novel insights and a strong foundation for future research, but would benefit from some revision of the present edition.
- The discussion should more explicitly address the study's limitations, particularly the lack of experimental (in vivo or in vitro) validation and the absence of data from patients with both Sjögren’s syndrome and periodontitis. Including these points will provide a more balanced perspective on the findings.
- Expand on the potential clinical implications of identifying CCR1 as a biomarker, including how it could be used in diagnosis or therapy, and discuss any barriers to clinical translation.
- Provide more detailed information regarding the preprocessing of the datasets, especially on how batch effects were handled and how data normalization was performed across different cohorts.
- Explain the rationale for choosing specific cut-off values (e.g., fold change, p-value thresholds) in the DEGs analysis to strengthen the transparency and reproducibility of the approach.
5.If possible, suggest incorporating preliminary experimental validation or, at minimum, provide a more detailed plan for future experimental studies to support the bioinformatics findings.
- There are a few grammatical and stylistic errors throughout the manuscript. A careful language revision is recommended to improve readability and professionalism.
Author Response
Comment 1: The discussion should more explicitly address the study's limitations, particularly the lack of experimental (in vivo or in vitro) validation and the absence of data from patients with both Sjögren’s syndrome and periodontitis. Including these points will provide a more balanced perspective on the findings.
Response 1: Thank you for your suggestions. The revisions are as follows: It is important to acknowledge the limitations of our study. First, the lack of in vitro/in vivo experimental validation (e.g., cell-based assays, animal models) precludes direct verification of bioinformatics-predicted molecular mechanisms, gene regulatory networks, and pathway functions. Consequently, findings remain correlational, impeding causal inference and mechanistic insights. Second, reliance on public database data introduces risks of sample heterogeneity and bias, potentially leading to overfitting or false positives in the absence of experimental validation. Third, the absence of preclinical validation limits the direct applicability of bioinformatics findings to clinical practice, confining conclusions to theoretical frameworks. Fourth, missing data on patients with comorbid SS and PD hinders the assessment of synergistic pathogenic mechanisms and immune microenvironment interactions. This gap undermines understanding of disease complexity in real-world settings.
Comment 2: Expand on the potential clinical implications of identifying CCR1 as a biomarker, including how it could be used in diagnosis or therapy, and discuss any barriers to clinical translation.
Response 2: Thank you for your suggestions. We have expanded the discussion on potential clinical implications and barriers to clinical translation in the last paragraph of the Discussion section. The revisions are as follows: CCR1 holds potential as an early diagnostic biomarker and longitudinal predictor of disease progression/relapse in SS and PD. CCR1 antagonists (e.g., BX471, CCX354) may block CCR1-mediated chemotaxis of immune cells (e.g., monocytes, T cells) to inflamed tissues in SS and PD. However, CCR1 shares homology with other chemokine receptors (e.g., CCR5), risking off-target effects. Developing highly selective antagonists remains challenging. Systemic CCR1 blockade may impair normal immune surveillance, increasing susceptibility to infections or malignancies. Biomarker-guided patient stratification (e.g., CCR1-high subgroups) could mitigate these risks.
Comment 3: Provide more detailed information regarding the preprocessing of the datasets, especially on how batch effects were handled and how data normalization was performed across different cohorts.
Response 3: Thank you for your suggestions. Batches were documented in each dataset’s metadata. For batch effect correction of gene expression data, we applied the ComBat algorithm implemented in the R package sva. Key parameters included: mod = model.matrix(~ disease_status), adjusting for biological variables. batch = batch_factor, where batch_factor represented technical batches. Batch effects were verified via UMAP visualization. Background correction was performed using rma() function in limma (log2 transformation + median polish). Quantile normalization was applied to align distribution shapes, followed by loess regression to correct for technical biase.
Comment 4: Explain the rationale for choosing specific cut-off values (e.g., fold change, p-value thresholds) in the DEGs analysis to strengthen the transparency and reproducibility of the approach.
Response 4: Thank you for your suggestions. FC>1.2 denotes a ≥20% gene expression change. This threshold is widely adopted to identify "moderate" expression differences, balancing the need to avoid missing potentially functional genes (with overly strict thresholds like FC>2) and to exclude biologically trivial fluctuations (with overly lenient thresholds like FC>1.1). For large sample sizes (e.g., n≥10 per group), a lower FC threshold (e.g., 1.2) can detect true differences by virtue of high statistical power. An adjusted p-value (adj p) <0.05 is a statistical standard for controlling false positive rates. Given that gene expression analyses often involve testing tens of thousands of genes, direct use of raw p-values leads to excessive false positives (e.g., unadjusted p<0.05 would misidentify ~500 of 10,000 genes as differentially expressed). Adj p (e.g., FDR corrected via Benjamini-Hochberg) restricts false positive rates to ≤5%, ensuring result reliability. The FC>1.2 and adj p<0.05 combination balances "statistical significance" and "biological relevance," making it suitable for most exploratory transcriptomic analyses.
Comment 5: If possible, suggest incorporating preliminary experimental validation or, at minimum, provide a more detailed plan for future experimental studies to support the bioinformatics findings.
Response 5: Thank you for your suggestions. We have added specific plans for future experimental studies in the last paragraph of the Discussion section. The revisions are as follows: Future studies will (1) integrate genomic/proteomic/epigenetic data to refine patient subsets and combinatorial targets; (2) validate differential genes in an SS-PD comorbidity model (NOD/Ltj mice with periodontal ligation) via therapeutic intervention and mechanistic assays; and (3) conduct a bi-omarker-guided clinical trial in SS-PD patients, integrating subgroup analysis of sensitive genes with CCR1-targeted therapies, following bioinformatics validation via cellular/molecular assays.
Comment 6: There are a few grammatical and stylistic errors throughout the manuscript. A careful language revision is recommended to improve readability and professionalism.
Response 6: Thank you for your suggestions. We have used Grammarly to correct grammatical and stylistic errors in the manuscript. The revised version has also been reviewed by native English-speaking scholars, who reported no significant reading obstacles.
Reviewer 3 Report
Comments and Suggestions for Authors
After reading your manuscript, I would like to compliment you on your topic choice. The subject is fresh and intriguing, and I think many practitioners will find your research helpful. The material is well structured and organized and the research is easy to follow. The results are presented in quality graphics and are accompanied by relevant explanations. The discussions are on topic and address recent and relevant references. The conclusions are clear and concise. The references are current, many titles in the last 5 years.
I would recommend:
- Add 2 more keywords to make them more relevant for the article
- Review the references once again and correct the writing style (some have pages, some do not, some have extra spaces before the volume, etc.)
Author Response
Comment 1: Add 2 more keywords to make them more relevant for the article
Response 1: Thank you for your suggestions. We have added common genes and immune cell infiltration as keywords.
Comment 2: Review the references once again and correct the writing style (some have pages, some do not, some have extra spaces before the volume, etc.)
Response 2: Thank you for your suggestions. We have corrected the mistakes of reference pages and extra spaces before the volume. Some journals use special pagination systems for typesetting convenience or digital publishing needs. For example, in some open-access journals, each article in the same volume starts page numbering from 0 and is distinguished by an article number, so citations may show only the article number resembling a page number. Other journals assign each article a unique numeric or alphanumeric identifier, which appears as a single-page number in citations. Many such journals in our references result in citations that seem to have only a single page number. Additionally, some journals use the publication year as the volume number, in which case the volume number is not separately marked. Additionally, some journals with special publication cycles do not display issue numbers. Therefore, the volume or issue numbers of some citations in this study are missing.
Round 2
Reviewer 1 Report
Comments and Suggestions for Authors
Dear Esteemed Authors, thank you for revising the manuscript accordingly. I appreciate the time and effort you put into my considerations. I would recommend that the Erudite Editors accept this manuscript for publication.